# Transdisciplinary Learning Communities to Involve Vulnerable Social Groups in Solving Complex Water-Related Problems in Bolivia

**Afnan Agramont [1],\*, Marc Craps [2], Melina Balderrama [3] and Marijke Huysmans [4]**

1   Universidad Católica Boliviana San Pablo and Vrije Universiteit Brussel, La Paz 15000, Bolivia
2   KU Leuven, 1000 Brussels, Belgium; marc.craps@kuleuven.be
3   Universidad Católica Boliviana San Pablo, La Paz 15000, Bolivia; i_balderrama_d@ucb.edu.bo
4   Vrije Universiteit Brussel, Department of Hydrology and Hydraulic Engineering, 1050 Brussels, Belgium; Marijke.Huysmans@vub.be
*   Correspondence: afnan.agramont@ucb.edu.bo; Tel.: +59-167-011-565

**Abstract:** Bolivia has influenced the international water arenas as a pioneer of the Human Water Rights Declaration before the United Nations General Council. However, despite a positive but rather ideological evolution, the country is still facing several water challenges in practice. Water governance is extremely complex due to intricate social structures, important spatial and temporal differences in the availability of water resources, ecological fragility, and weak institutions. A Transdisciplinary Learning Community approach has been adopted by the Universidad Católica Boliviana to take into account the complexity of the water problems caused by social, hydrological, and ecological system imbalances. In this approach, researchers and non-academic actors work closely together to integrate different ways of conceiving, using, valuing, and deciding on water issues. The approach aims at co-creating resilient solutions by recovering and restoring not only the ecological system, but also the social system in which all actors are aware of their role and responsibility. We explain the challenges and concerns raised by this approach in a case study of the Katari River Basin (KRB), which is impacted by a high degree of contamination that is mainly caused while crossing El Alto city, leading to dramatic consequences for the Lake Titicaca ecosystem and its surrounding communities.

**Keywords:** transdisciplinary learning; water governance; integrated water resource management; collaborative learning; knowledge co-production; river basin management policy

## 1. Introduction

Water is becoming a scarce resource with difficult access in many places. For more than 844 million people in the world, basic drinking water is still a dream, and for another 2.3 billion people who lack basic sanitation services, the Millennium Development Goals on water and sanitation have not been fulfilled [1]. According to the World Health Organization (WHO), diseases associated with the lack of drinking water, adequate sanitation, and hygiene are still one of the main causes of mortality for millions of inhabitants of developing countries [2]. Almost half of the inhabitants of developing countries suffer from diseases that are caused directly or indirectly by the consumption of polluted water or food, or by organisms that cause diseases that develop in water.

Besides, half of the water in drinking water supply systems in developing countries is lost through leaks, a lack of maintenance, poor dimensioning, illicit connections, and/or vandalism. As the population grows and income increases, more water is needed. In the year 2025, demand for water will have increased by at least 50% compared to 1995 [3]. As a consequence, water has become a crucial element for socio-economic development. Water is such a necessary resource that its infrastructure

(dams, irrigation channels, purification and desalination technologies, sewage systems, and wastewater treatment) could become the object of political fights, social conflicts, and international wars.

The Sustainable Development Goals (SDGs), which were adopted by the United Nations in 2015 as part of the 2030 Agenda for Sustainable Development, place even greater demands on societies than the MDGs, which they replace [4,5]. Addressing water provision for all (SDG 6), while taking into account interactions and trade-offs with other global challenges such as climate change, renewable energy, food and health, requires strong institutions (SDG 16) as well as partnerships between public, private, and civil organizations (SDG 17) [6]. Scientists of different disciplines will have to design metrics, establish monitoring mechanisms, and propose adequate criteria in close cooperation with all involved actors [7]. Within SDG 6, SDG 6.5 aims at the implementation of Integrated Water Resource Management (IWRM) as part of the response to "ensure [the] availability and sustainable management of water and sanitation for all" [6].

IWRM is internationally recognized as the course to make an efficient, equitable, and sustainable use of water resources [8]. This approach tends to reform institutional arrangements and systems of water governance [9,10] under the premise that the participation of actors and sectors at all levels is essential to achieve sustainability [8]. By definition, the IWRM requires a high level of institutional coordination and collaboration among stakeholders. A collaborative system of governance among governmental and non-governmental actors, organizations, and sectors facilitates a more efficient, effective, and equitable management of the water resources [10]. This demands the interaction between individuals belonging to diverse disciplines, backgrounds, and cultural realities.

In this article, we want to focus precisely on the interactions between scientists of different disciplines and representatives of different stakeholder groups in water management. To deal adequately with the wicked character of the socio-hydro-ecological system, new transdisciplinary research practices have to be developed that are able to meaningfully link different insights, interests, and values. In the Global South, socio-economic inequalities cultural differences, and the social exclusion of native communities and other vulnerable social groups often aggravate the challenges for transdisciplinary research. However, at the same time, we find there a fertile soil for inspiring alternative research practices to bridge the gap between sciences and society.

In what follows we analyze the ongoing experience with Transdisciplinary Learning Communities (TLC) initiated by the Universidad Católica Boliviana (UCB) in Bolivia to address the complex problems of water quantity and quality that affect the Katari River Basin, flowing into the Lake Titicaca. In this case, the main question is how a transdisciplinary approach can contribute to taking the interlinkages between the different ways of making sense and dealing with water-related problems into account.

After this introduction, we situate our research question in the broader framework of IWRM and the need for collaborative water governance. Then, we describe the TLC approach as an action research methodology in our case study of the Katari River Basin. In the results section, we first describe how water management has evolved in recent years in Bolivia; then, we analyze the current situation in the KRB with the TLC approach. In the discussion section, we look more in depth at the challenges with which the TLC is confronted. In the final conclusions, we look forward to how our current experiences can inspire future actions, not only in the KRB but in many other river basins with similar characteristics in the Andes and elsewhere in the Global South.

## 2. Integrated Water Resources Management and the Need for Collaborative Governance

Water resource management has played a historical role in nation-building worldwide [11]. The International Conference of Water and Environment in Dublin in 1992 defined four principles as the core of water resources management. First, water is a finite and vulnerable resource that sustains life, the environment, and economic development. Second, water management should incorporate participation of all sectors, users, planners, and policy makers, at all levels. Third, women should be considered as having an essential role in water management. Fourth, water holds an economic

value and should be considered an economic good based on all the competing uses of this natural resource [12].

In 2000, the discussions continued at the Second World Water Forum in The Hague, in which the statement "a water crisis is often a governance crisis" reflected the need of integrated institutional and managerial arrangements as prerequisites to finding sustainable solutions to problems related to the water sector [13]. This international conference incorporated the concept of Integrated Water Resources Management (IWRM) in the international development agenda in order to favor a holistic view of the water sector, and promote it as the approach to find sustainable solutions for problems related to the water sector.

The definition provided by the Global Water Partnership (GWP) in 2000 stated, "Integrated water resources management is based on the equitable and efficient management and sustainable use of water and recognizes that water is an integral part of the ecosystem, a natural resource, and a social and economic good, whose quantity and quality determine the nature of its utilization". This approach gained attention and spread worldwide based on three principles that the definition incorporated. First, water management should consider this resource as essential for economic development. Water is used in a diversity of productive sectors, such as industry, mining and agriculture, relying on its availability and quality. Second, water is vital for social systems. A just and fair distribution between uses of water allows societies to avoid problems of social conflicts and public health that are related to water availability and quality deterioration [14]. Third, ecosystems rely on water resources (quality and quantity) to survive and keep a good ecological status.

At the same time, IWRM delineates the river basin as the best biophysical management boundary due to the ecological, social, hydrological, and institutional relations embraced within this system [11], as upstream water activities, practices, and values have a direct influence on downstream environmental, ecological, and social systems. Furthermore, this element aims to connect actors and sectors across the river basin, promoting the participation of stakeholders in decision making as one of its main pillars [8,11], in order to find a balance between the sector and actors' water interests.

However, the three principles incorporated in the IWRM approach tend to be contested, since water uses compete for the allocation of water resources based on sectorial interests. Moreover, productive sectors such as agriculture and mining, which have positive impacts on local economies, heavily impact the quality of water resources and their ecological status. To reach sustainable development, which according to the now classical definition of the 1987 Brundlandt report, is "development that meets the needs of the present without compromising the abilities of future generations to meet their own needs", IWRM must take into account economic feasibility, social equity, and environmental issues. Moreover, since the time of origin of that definition, more emphasis has been put on ecological boundaries and social inclusiveness [15], which is linked to new forms of water governance [9]. For that purpose, the IWRM approach facilitates sustainable management by considering the river basin as an appropriate natural boundary that connects the actors, sectors, and up/downstream relations. At the same time, IWRM considers the participation of all stakeholders in decision making as one of its main pillars [8,11]. As water governance refers to the interplay between political, social, economic, and administrative systems to manage and develop water resources and to deliver water-related services [16], the IWRM approach is nowadays commonly considered as the most influential water governance model worldwide [17].

Moss called attention to the necessity of appropriate sectorial interplay to enhance sustainability in water resources management, since water-related issues are often also related to other policy sectors that influence or rely on the water sector [18]. Additionally, the vertical interplay presented by Moss reflects the need for a co-action across administrative levels to face problems of scale linked to river basin resource management [19] in order to organize institutional arrangements, from national administrative levels to local levels.

However, although IWRM is widely accepted as the best water management approach worldwide, the implementation is still a challenge in diverse contexts [8,12,17,20–29]. Moss highlighted that the

'successful' examples of IWRM worldwide reflect diverse water problems and the objectives behind their solutions [18]. The implementation of solutions in the United States is linked to navigation, energy production, and flood control [30,31]. In Spain, the implementation of IWRM principles is connected to agricultural reform [32]. In England, the employment of IWRM shifted from flood control to ensuring water services for the urban population [33]. For instance, the Tennessee Valley Authority in the United States is a recalled positive example of IWRM application [8], since the results experienced by the adoption of the Tennessee River Basin Management improved energy production, navigation, flood control, and agriculture [31] through a participatory process of decision making [30]. Consequently, the success of an IWRM policy implementation heavily depends on the sensitivity toward the local context, the understanding of existing governance arrangements, and the specific objectives linked to the local identified water problems [34].

Failure to consider the (social) embeddedness of natural resources management has been a key factor in limiting the success of previous governance reforms. For that reason, various scholars already argued more than a decade ago for collaborative water governance and social learning to broaden the IWRM scope of administrative coordination [10,35–37]. Indeed, water-related problems are typical examples of "wicked problems", which cannot be objectively described nor definitively solved [38], as multiple stakeholders with different perspectives and possibly conflicting interests are all involved in it. They are intertwined with other wicked problems such as climate change, decreasing biodiversity, extreme poverty, and migration flows. They are characterized by a plurality of decision makers, pervasive uncertainties and ambiguities, spatial and intertemporal externalities, an interplay of human and natural components, and an evolving understanding of policy objectives. They are not neutral objects of inquiry, but already from the problem definition they are value-laden and guided by a perspective toward a more desirable state of affairs [39].

When wicked problems are at stake, collaborative governance has been widely recommended to replace top–down and technocratic approaches. This means that all relevant stakeholders are actively involved in the development of water policies to reconcile environmental, economic, and societal goals [40]. Governance results then from an interaction process between different actors confronted with a shared problem, in the search for synergetic solutions through the joint appreciation of different but complementary viewpoints [41–43]. Through an emergent and possibly conflictive process, actors increase their insights in the intertwined nature of complex problems while they negotiate mutually beneficial agreements.

Barbara Gray [41,42] described this joint problem-solving and decision-making process in four phases: problem-setting, direction setting, implementation, and institutionalization. Critical tasks in the first problem-setting phase include identifying the relevant partners and getting them to commit to a collaborative partnership. Then, direction setting implies the exploration of shared issues and reaching agreements about how to address them. The implementation of a collaborative initiative has to engage the involved actors in the execution and follow up of the agreements. Finally, the institutionalization of collaborative governance has to structure and regularize the ongoing interactions among the actors, and it has to enhance social learning for continuous adaptations and improvements, and for the replication of similar partnership processes in other contexts.

However, it is difficult for this collaborative governance approach to be put into the practice of river basin management [10]. The process might be quite demanding for the participants, as they have to be able to reflect on their—often implicit—assumptions concerning their views about how (water) problems have to be managed, and analyze how different views affect each other and can be linked constructively to foster a shared vision. This implies that the knowledge generated by the water specialists has to be linked with the knowledge of their colleagues from other (human, social, economic, environmental . . . ) disciplines and with the information, insights, and values of governmental, private, and civil stakeholders. Reflexivity allows questioning values, background assumptions, and normative orientations with four purposes in the search for sustainability: (1) to develop a shared understanding of a problem, (2) to reflect on the social relevance of the problem framing, (3) to set up joint social

experiments and collective learning processes between the involved actors, and (4) to create a critical research agenda that is able to help transform the current governance system into a more sustainable system [44].

Due to the importance of the quality of the relations between the actors that participate in collaborative water governance, the attention is then drawn to activities that enable the development of joint knowledge, such as: getting the attention of all stakeholders and raising awareness of their mutual dependency, getting their commitment to engage in a joint learning endeavor, legitimating participants and leadership, connecting stakes and interests, dialoguing to explore diverse views and action possibilities, negotiating roles and contributions, guaranteeing the commitment of constituencies, and aligning efforts and agreements [45,46]. Although these activities may not be considered independent from the more conceptual, technical, and legal governance issues, they need specific attention and require specific competencies and skills that can be acquired through social learning [36,37].

In the next section, we explain how the transdisciplinary learning approach that was applied in our case study aims at stimulating social learning between researchers, public authorities, and other stakeholders, with special attention for the vulnerable local communities, which is necessary for collaborative water governance.

## 3. Transdisciplinary Learning Communities as Action Research Methodology

The water research project to which this article relates is part of a broader Transdisciplinary research program between the Universidad Católica Boliviana San Pablo (UCB) in Bolivia and the Flemish Interuniversity Council (VLIR-UOS). With this program, the UCB wants to enhance its social impact by giving priority in research and outreach to the most vulnerable social groups in the Bolivian society. The lack of capacities of the UCB to counter the challenges presented by the intertwined problems of water quality and quantity, food sovereignty, social discrimination, legal rights, and a lack of economic opportunities that affect the poorest communities is common in the higher education landscape in Bolivia and in the Global South more in general. Universities invest most of their efforts in teaching, with a subsequent lack of engagement with society´s development. To reverse this situation, universities not only need to strengthen their relations with national and international research networks to increase the academic quality of their research. They also need mechanisms to co-create, together with local actors, knowhow that is adapted to the local socio-ecological and cultural circumstances of the most vulnerable groups. For this purpose, the university does not only need resources and technical knowledge for doing sound scientific research, it also needs to become more acquainted with the reality as it is experienced by the vulnerable groups that the university intends to support. The university also needs acceptance as a legitimate partner by the communities and other developmental actors. However, the UCB has a limited tradition and experience with this kind of research. Spaces and opportunities for dialogue between academic scholars and developmental actors are scarce. Cultural differences, disciplinary and institutional boundaries, and social inequalities complicate an open, bilateral exchange of information. Women, adolescents, and children are often not involved in the development of solutions for their own problems.

To reach its strategic objective to contribute substantially to the improvement of the living conditions of the most vulnerable groups in Bolivia, the UCB started a long term inter-institutional cooperation program with the Flemish Interuniversity Council (VLIR-UOS), which has a long tradition in academic development cooperation (www.vliruos.be/en). Its motto, "Sharing minds, changing lives", expresses its mission of supporting partnerships between universities in Flanders and the Global South, to transform them as drivers of local and global sustainable development. Although the VLIR-UOS cooperation receives positive evaluations in general, there is need to better assess the consequences of the predominantly disciplinary and expert-driven way of doing research on the sustainability and social impact of the results. This resulted in a group of Flemish and Bolivian academics who were motivated to put a collaborative and transdisciplinary learning community

approach at the core of the program with the UCB. The vision of the program is that of a community of academic scholars, belonging to different disciplinary projects, who work closely together with each other, with public and non-governmental development actors and with local communities in so-called "Transdisciplinary Learning Communities" (TLC). Together, the participants in the TLCs aim at co-creating "actionable knowledge" [47], which is adequate, contextualized, and "ready to use" by the local actors, because by participating in the research, they develop the knowledge, competencies, and willingness to put the results in policies and practices.

Transdisciplinarity refers to integrating different forms of knowledge from different academic disciplines and from different social actors (policy makers, local communities, non-governmental organizations, companies . . . ) in a joint knowledge production process [48]. The concept of learning communities is inspired by the situated learning theory of Lave and Wenger [49]. In their conceptualization, communities do not refer to homogenous social or ethnic groups, but rather to emergent and informal groups of people—crossing the boundaries in and between existing organizations—that engage in shared efforts for collective learning [50]. In this way, TLCs aim at addressing the complex nature of the problems with which the most vulnerable social groups are confronted, intertwining water management and governance with issues of food sovereignty and nutrition, production development, human and indigenous rights, and social conflicts.

According to a transdisciplinary view, different disciplines are not dealing with different parts of reality; instead, they offer different frames to make sense of the real world built upon experiences. Consequently, a river basin is not only a physical–hydrological phenomenon, it is also a socio-geographical system, with legal–administrative implications within which an economic dimension is also incorporated with a variety of productive activities depending on or impacting the water quantity and quality. At the same time, it is a place of living with a sense of belonging for families and communities. Considering a river basin as a complex socio-economic and ecological system also draws the attention to the interrelations between a variety of social groups, living in rural, peri-urban, and urban environments with different cultures, interests, and expectations.

Through the TLCs, the university is building networks beyond its own walls to engage in and benefit from system-wide collaborative learning. The collaborative approach of the TLCs offers space for exchange and dialogue between the university, local communities, governmental agencies, non-governmental organizations (NGOs), and other participants. However, as there is a limited tradition of doing this kind of research in Bolivia, and since the legitimacy of the UCB is still under development, the necessary involvement of the other actors in the TLCs is far from evident. National public agencies demonstrated only a limited interest to be part of it during the preparation of the program. Apparently, they also often lack knowledge about problems from the perspective of the most vulnerable social groups. Nevertheless, their participation is important, and is sought for through formal agreements. Local public (municipal and departmental) agencies showed a stronger interest and greater disposition than the national agencies for getting involved in the TLCs, which can enable access to the local communities. NGOs and indigenous organizations also showed a high interest in the program, even without having clear ideas on how their members were going to benefit from it. For that purpose, a permanent dialogue with them about how the TLCs can favor the interests of their members is a key condition for the success of the TLCs.

A collaborative research strategy is considered most adequate to generate the "actionable knowledge" that takes into account the specific needs, values, and interests of the most vulnerable social groups in the river basin. This means that researchers from different disciplines work closely together with the stakeholders in their study throughout the whole research process. There is a long tradition of Participatory Action Research (PAR) in Latin America, which can be built upon. However, these PAR methodologies have been predominantly developed outside the universities by non-governmental organizations and social movements. As a consequence, the emphasis is often more on social action, and they are poorly integrated in the academic world, and are predominantly in the social sciences, if their presence is felt at all. The current tendency of Latin American universities to

focus more on academic research risks marginalizes these PAR methodologies even more as a form of "social outreach" that is separate from the real scientific research. That is the reason why we focus here on the collaborative research methodology as a means for generating knowledge that is relevant beyond the specific case in which the research was done. Although this kind of research is realized in such a way that the contextual factors are fully accounted for, it does not lower the academic quality requirements, nor does it mean that the results are only relevant for the case under study. It informs theory that is applicable in other cases as well [51]. The "thick description" [52] of the KRB case allows an "inferential generalization" of the conclusions by considering carefully the contextual similarities and differences between this case and other cases in Bolivia or elsewhere, and then adapting the conclusions accordingly. The learning conclusions of this specific case are also compared with the insights of existing theories regarding the implementation of IWRM. By confirming or disconfirming the existing theories through the evidence in this case, the study also allows so-called "theoretical generalization".

The KRB, see Figure 1, was identified as a convenient case to study collaborative water governance. It presents an extremely high degree of ecological and social vulnerability, with tensions and conflicts between upstream and downstream, and between urban and rural areas. In the following paragraphs, we describe only briefly the geographical and hydrological basic facts and figures of the KRB. In the next section, we analyze more in depth how the recent socio-economic history and water policies in Bolivia have affected the water quality and quantity, and how this has impacted the living conditions of its inhabitants.

The KRB is located in the Andes region near La Paz, which is the capital of Bolivia in South America. This river basin covers 2955 km$^2$, and its altitude ranges from 3800 to 5720 meters above sea level, with slopes from 4% to 90% [53]. The mean temperatures in the last 15 years have oscillated between −5 °C and 15 °C, and the average precipitation ranges from 470 to 742 mm [54]. The average river basin outflow at the Cohana Bay, the discharge point, is 7.7 m$^3$/s [55].

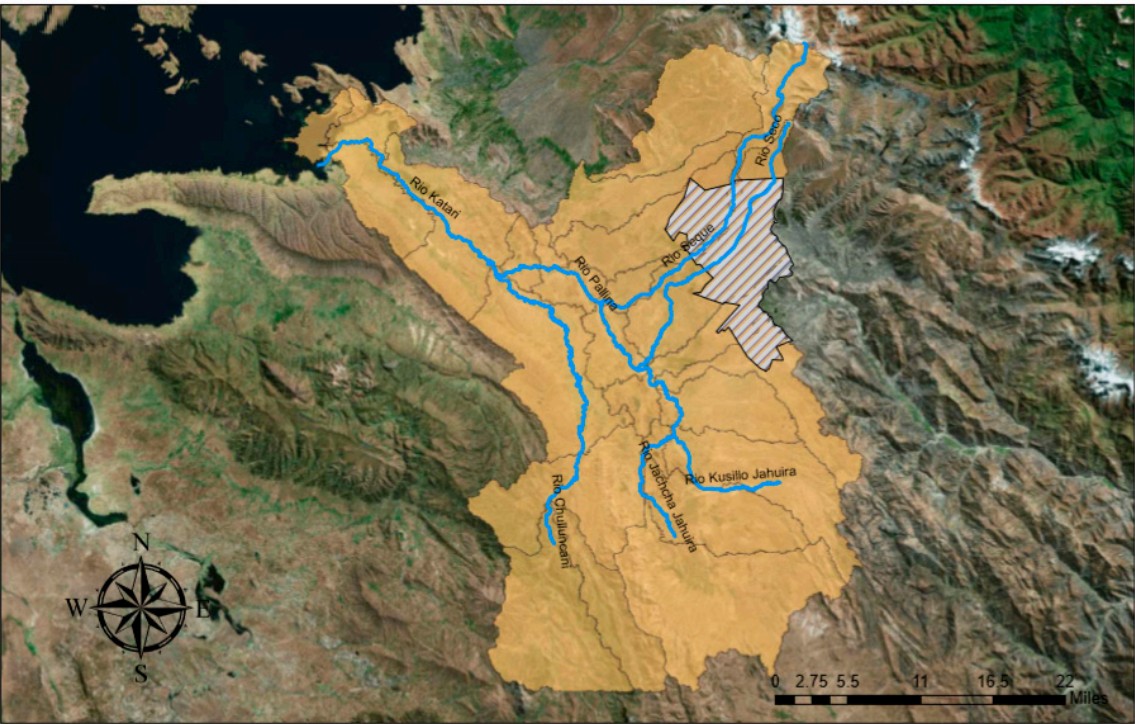

**Figure 1.** Katari River Basin.

The KRB is composed by the following rivers: at the highest altitude, the river basin is composed by the Seco and Seke rivers. These two rivers flow in the Pallina river, which later joins the Katari river, and finally discharges at the Cohana Bay into Lake Titicaca.

The KRB with over one million inhabitants is the most populated river basin in Bolivia [56]. The region of the Katari Basin has one of the highest rates of population growth in the world, which is due to massive rural migration to the city of El Alto and its surroundings [57]. The river basin crosses the city of El Alto, which is known as one of the fastest-growing regions in the world [57]. El Alto experienced an increase of 95,000 to 1.2 million inhabitants between 1976–2011 [58]. This massive population increase has contributed to environmental problems related to water resources in the region.

## 4. Results

We start this section with an analysis of how water policies have evolved in Bolivia and how this has impacted the socio-ecological conditions in the river basins in recent history; then, we analyze more in depth the challenges for collaborative water governance in the Katari River Basin.

### 4.1. Integrated Water Management and the Challenge of Collaborative Governance in Bolivia

In Bolivia, water governance has struggled intensively with the wicked interplay of social, ecological, and economic factors that go beyond the conventional scope of technically informed management. To address this wickedness, the Bolivian National Government followed the global tendency of incorporating the IWRM approach in its national policy agenda as the mechanism to face water problems at the national level in 2006 [59]. The policy was buttressed by the creation of a Vice-Ministry of Water Resources and Irrigation, and was materialized by the creation of the National River Basin Plan, with the aim of the "good" governance of water resources in Bolivia [60].

Despite the incorporation of international policy mechanisms at the national level, water governance challenges are still increasing in Bolivia due to political, administrative, economic, and social factors. Inefficient water services, asymmetries in resource allocation, ambiguity of the regulatory framework, lack of control, and unclear water rights continue to be part of reality. In some cases, these conditions result in significant social conflicts, which are experienced for instance during the so-called "Cochabamba water war" between the local inhabitants and its government, which was willing to privatize the drinking water service in early 2000.

The Bolivian water governance reflects a lack of coordination between subnational and national authorities, which is mostly due to political differences between both levels. As a result, there is an evident inefficient vertical interplay [19,61] that heavily influences the design and implementation of water policies. Moreover, the design and implementation of the water policy depends on the highest level of the administrative system, the National Government, since its technical executive department is considered to hold the expertise in the water field. As a result, national policies neglect local knowledge as crucial for sustainable solutions for water resource management.

In Bolivia, there is a tendency to have boundaries between policy sectors [62], which results in obstacles for the necessary sectorial interplay [61,63]. As a matter of fact, other sectors such as agriculture, energy, and mining heavily influence water resources management. In the last decade, Bolivia has experienced a growing trend in its GDP [64], which is linked to the increase of production in various sectors, and consequently, there is also a growing water demand. For instance, from 2006 to 2015, there has been an increase of 120% in agricultural production, which requires a significant volume of water resources to irrigate the 3.68 million hectares occupied by this sector [65]. At the same time, the mining production increased 205% for the period 2006–2015 [66], which has extraction and refinery processes that also demand water resources. The growing economy also has a link with water contamination, as increasing amounts of the solid waste of industrial and urban pollutants are discharged in the water courses.

The coordination, participation, and collaboration among local administrative actors regarding decision making and the planning of water resources present serious challenges in Bolivia. The country

holds a complex social structure with a growing power asymmetry between the urban and rural areas. This power asymmetry results in asymmetries in water allocation and water contamination, which often has the biggest negative impact on the most vulnerable communities. Furthermore, water services heavily rely on national authorities, which do not always understand local realities, and as a result, potential water conflicts tend to be reproduced.

Conflicts can find their origin in claims of limits between communities, municipalities, or departments (because water sources or water courses are disputed by those who claim property), the wastewater treatment plant allocation, and water services' fees. To all these potential reasons for conflicts, the increasing demand of water for the generation of electrical energy must be added. It is indeed the goal of the national government to guarantee 100% coverage of electricity, clean water, and sanitation by 2025 for all Bolivian citizens, which will rely on the availability and quality of water resources.

At the same time, the Bolivian water regulatory framework does not support sustainable water resources management. The Bolivian water law was adopted in the early 2000s. This law requires an update to align it with the legal system, which regulates the water resources and services. The Bolivian water law of 2000 recognizes the participation of the private sector in water services management. However, the 2009 new Constitution of the Pluractional State of Bolivia neglects the participation of the private sector in water management issues.

There is an ambiguous definition of the water management roles, and/or the formal roles are usually not followed. Administrative responsibilities between different public entities at national and local levels tend to overlap, while key issues such as water withdraw and discharge permits are not covered by any authority. Due to the lack of water abstraction and discharge registration, there is no clear information for the planning and application of a regulation that puts into perspective the availability of water and the future responsible uses of it, taking into account the sustainability of water as a delicate natural resource. On the other hand, under the national constitution, there is a clear definition of the responsibility for the water services at the urban level. Here, the municipal government holds the main responsibility. However, in the main cities, as in La Paz, the National Government attributes to itself the responsibility for the management of the water company. In the city of Santa Cruz, the urban water provision relies on a water cooperative whose shares are owned by the water users connected to the system.

There are no public records of water allocation, which is supposed to be acquired through concessions or legal provisions. Such a water abstraction and discharge registration could then assign rights to the use of certain volumes of water in such a way that it does not put ecological and hydrological balances at risk, by taking into account the projection of the demand in the future and the return of part of the water (treated and in good conditions) to the same source from where it was taken in a sustainable way.

Currently, the appropriation of water resources occurs in two main ways: by territorial appropriation (the purchase of land, including its natural water resources and biodiversity), and by the "official concession" of water. In Bolivia, a large portion of water resources is allocated to indigenous communities. They consequently grant the legal right over these water resources under a territorial jurisdiction framework. However, local communities are not included in the decision process to promote a dialogue about water governance and the implications for local development.

*4.2. Challenges for Collaborative Water Governance in the Katari River Basin*

The hydrological system of Katari gained attention in 2004 when the president at the time promulgated the law 2798, declaring this river basin "an environmental disaster zone and hydrological emergency" due to the contamination resulting from the anthropogenic activities that had developed within its main tributaries. The reason for this law has to be sought in the connection with Lake Titicaca, which is the largest freshwater lake in South America and the most relevant water resource

body in the Andes Region, into which the Katari river discharges [67]. Lake Titicaca is under the binational water management of Bolivia and Peru.

The environmental evaluation performed by the Bolivian Vice-Ministry of Water Resources and River Basins identified two main problems related to the water resources in the Katari Sub-Basin. First, the increasing population has led to the establishment of factories, mining activities, tanneries, slaughterhouses, and agriculture, which all put a large amount of pressure on water resources demand. Second, the mentioned activities have also damaged the water quality of the sub-basin, since environmental legislation is not enforced [58]. Only 55% of the residential sewage is processed at a wastewater treatment plant, and the effluents of these water treatment plants do not comply with current water quality regulations. Additionally, there are problems with solid waste management in the municipalities within the KRB [68], which further negatively affects water quality.

As framed by Archundia [69], illustrated in Figure 2, there are four regions of anthropogenic influence on the water resources in the Katari River Basin: (1) Milluni valley, (2) El Alto, and Viacha, (3) the Katari rural lowlands, and (4) Lake Titicaca (see Figure 2). The Milluni sub-basin is situated at the highest location of the Katari River Basin. This sub-region, which is located at 4450 meters above sea level, is affected by the exploitation of tin, lead, and zinc, which was initiated in 1920 through a private enterprise named "Fabulosa Mine Consolidated", which was owned by British shareholders. Later, from 1975 to 1986, the Bolivian company COMSUR continued the operations. Currently, the mine is not under an official concession contract, but two local communities dispute the exploitation of the minerals when tin prices increase [70].

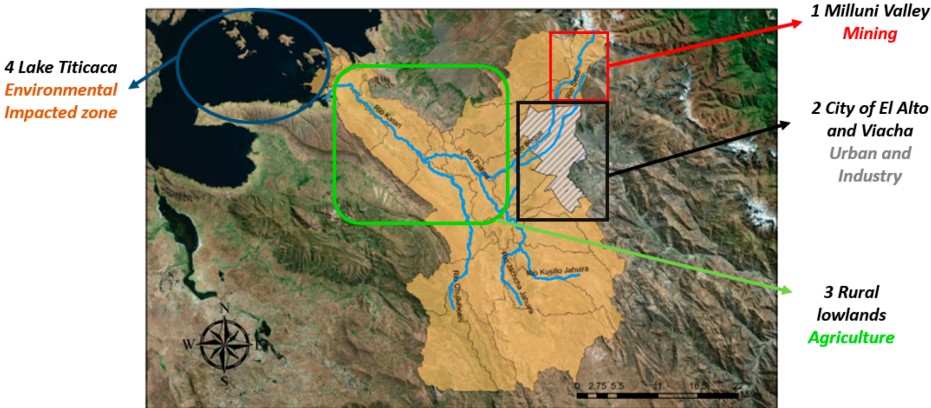

**Figure 2.** Katari River Basin/sub-regions of study.

Over 90 years of mining operations in the Milluni sub-basin have left a legacy of water contamination. Local geology holds a high concentration of sulfide minerals. When the minerals are exposed to the atmosphere (oxygen and water), which happens in conventional mining processes, they are a constant source of acid mine drainage. The low pH of the acid mine drainage enhances conditions for the incorporation of dissolved metals in the water. Downstream of the mining area, the second largest water reservoir is located, which is known as the Milluni Dam. This dam holds a capacity of 10 million cubic meters, and partially supplies water to the cities of El Alto and La Paz. The local drinking water supply company currently incorporates additional steps to treat the contaminated water in order to comply with local standards for drinking water.

Downstream of the Milluni valley, in the cities of El Alto and Viacha (Figure 2, Urban and Industrial zone), pH recovers to a neutral state due to a natural oxidation process during transportation. At these locations, the water of the Seco and Seke rivers, which are part of the Katari River Basin, is considered to be of good quality, according to Bolivian local standards. However, when these rivers cross the city of El Alto, their water quality heavily deteriorates. The accelerated massive urban migration of El Alto did not allow a proper city planning of the infrastructure for water services. The city of El Alto has only one wastewater treatment plant, which has an installed capacity of less

than 55% of the total effluents generated by the local population. Moreover, this wastewater treatment plant, which is known as the "Puchukollo", discharges an effluent of low water quality that cannot be employed for irrigation or any other purposes [69].

After crossing El Alto, the Seco and Seke rivers qualify as eutrophic waters due to the high concentration of nutrients and low dissolved oxygen concentration, which limits the presence of the endemic species that are characteristic of these types of ecosystem. The anthropogenic influence of the city of El Alto also incorporates fecal contamination to concentrations that are harmful to public health.

The third sub-region of study consists of rural communities whose livelihood depends on agriculture in the area. Their main activity is linked to livestock and small-scale diary production in the region. However, the problems of contamination caused by the cities upstream of El Alto and Viacha, which were mentioned before, do not appear to be the main problem identified by these communities. They claim to be historically neglected communities lacking water infrastructure to develop and increase agriculture in the region.

The last sub-region of the study incorporates the communities around the Cohana bay at Lake Titicaca. These communities are heavily impacted by the contamination developed upstream. However, since the promulgation of the law declaring the Katari river an ecological disaster, nothing has changed, and the river basin contamination is still impacting the environment and the local livelihood as before.

The approval of the law 2798 in 2004 by the Bolivian National Congress, declaring the Katari Sub-Basin an environmental disaster zone and an area of hydrologic emergency, meant that the national government recognized the seriousness of the environmental problems in the KRB, and that it declared environmental remediation to be a national priority. At the same time, the law granted the involved municipalities the possibility to restructure their annual operational budgets in order to implement plans, programs, and projects to improve the environmental scenario. Furthermore, the law gave origin to the "River Basin Environmental Conservation Management Committee" for the Katari River Basin. The governance model was structured under a board of directors that was composed by representatives of the national government, the association of municipalities, universities, the federation of farmer unions, neighborhood associations, and the water services company "Aguas del Illimani".

The enacted law of 2004 mentioned important principles for an integrated, transdisciplinary, and collaborative approach regarding the river basin. First, the law recognized the importance of a participatory and multi-sectorial approach to articulate actions for the improvement of the environmental scenario. Second, the law recognized the importance of establishing the hydrological boundaries of the river basin as the basis to delineate an administrative system.

However, although the enacted law seems to contain important principles for an integrated river basin governance, the "River Basin Environmental Conservation Management Committee" did not issue any plans, programs, or projects afterwards. One of the reasons of this lack of initiatives might be the ambiguity of the institutional framework resulting from this law. The law vertically designated a number of heterogeneous organizations to be part of the Management Committee, but without a process through which these actors could come to a joint understanding of the different problems and an agreement regarding the possible solutions and the contribution of each to these solutions. Moreover, the committee also lacked information and expertise to establish the adequate tools and mechanisms for the required operations.

Later, in 2010, the Bolivian National Government incorporated the IWRM approach in the national development and environmental agenda through the establishment of a new government agency called the "River Basin National Plan". This government agency developed various river basin plans with the objective of articulating the IWRM in the river basins prioritized by the national government; among these was the "KRB Director Plan", which was published by the National River Basin Plan in 2010.

The "KRB Director Plan" identified three dimensions that are in continuous contestation with each other: the environmental, the socio-economic dimension, and the institutional/administrative dimension. The plan presents a problem definition that incorporates problems related to water

quality and water quantity regarding the contamination of the Katari River and Cohana Bay, and the insufficient water resources to attend to all of the local demands. The plan also points to the diversity of the stakeholders and social sectors that are involved and the legal plurality in the area as special challenges for an integrated management of the Katari River Basin (IMKRB).

As shown in Figure 3, the "KRB Director Plan" structured a governance model consisting of four parts. First, the River Basin Board is composed of local authorities, as well as members of the state government, municipalities, the national irrigation service, and the vice ministry of water resources and irrigation. Second, an Executive Unit is responsible for the implementation and daily management of the plan. Third, a Technical Council is composed of "institutions and enterprises" with technical water management expertise. And last, a Participatory Forum "coordinates and communicates with the productive and social forces" present in the river basin.

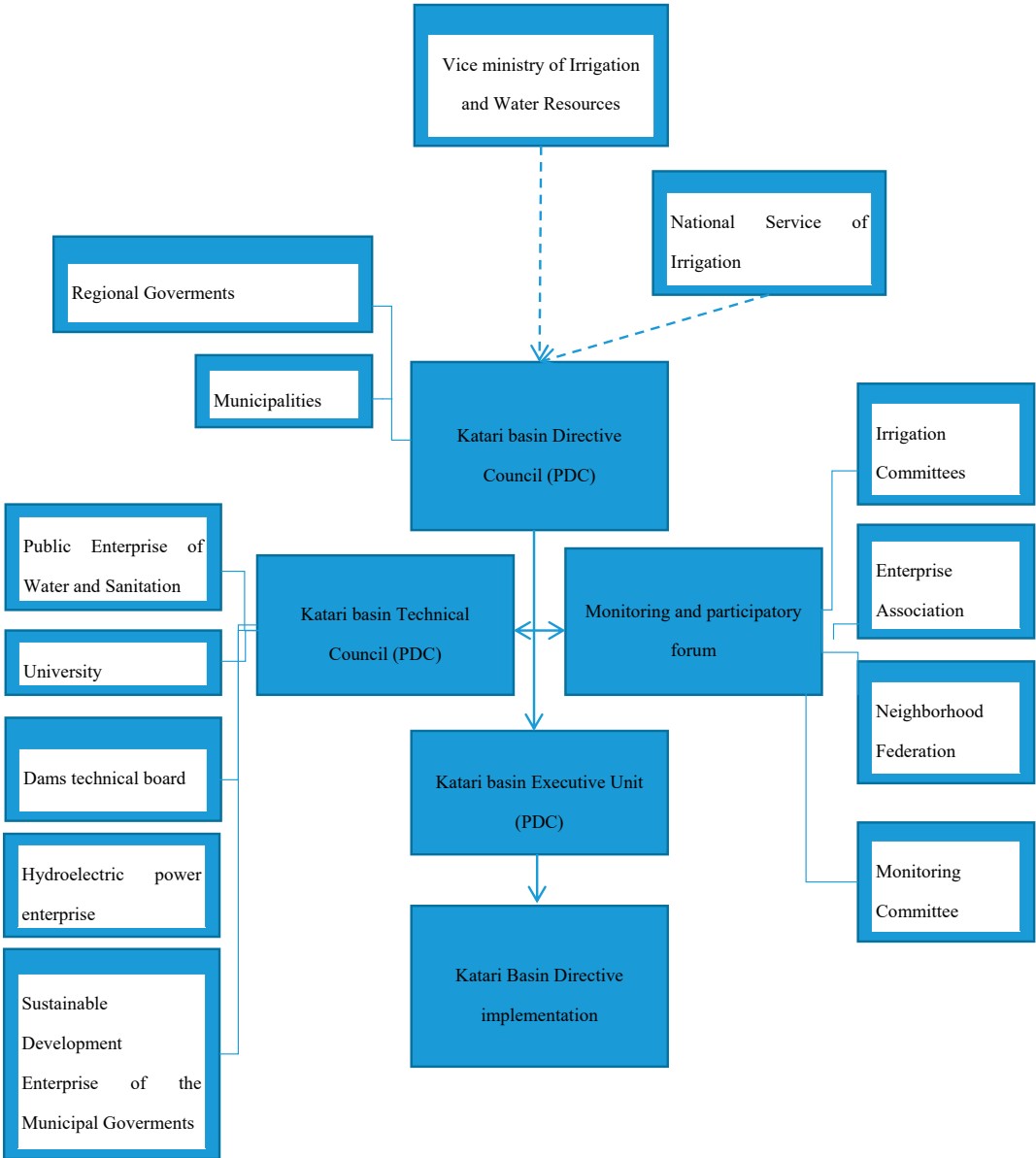

**Figure 3.** KRB Institutional Structure from 2010 Katari Basin Director Plan.

However, the river basin management plan had not been implemented until 2016, after the algae bloom experienced in 2015 at the Coahana Bay which resulted in two tons of dead fish collected at

the Titicaca shores. Furthermore, the 2010 Katari Basin Director Plan did not incorporate concrete measures to face the environmental problems.

As the KRB management plan currently under implementation was developed and implemented exclusively by the National Agency of the Ministry, there is a lack of participation by the municipal and community actors in the design and implementation of the KRB management plan. Consequently, the existing ambiguity concerning the causes, responsibilities, and possible solutions of the problems cannot be addressed adequately [24]. This decreases the legitimacy at the local level of the proposed solutions for the problems of the Katari River Basin.

## 5. Discussion

To counter this situation, the TLC initiated by the UCB helps provide a holistic view of the problems, as it offers a space in which academic experts with different disciplinary backgrounds as well as policy makers and local communities link their knowledge, experience, and expectations. To mitigate the influence of water pollution and water quality deterioration on the urban and rural vulnerable population (indigenous communities, slums, farmers' associations, and secondary cities settled along the Katari sub-basin), the development of a water quality monitoring strategy is considered to be a priority. This implies identifying the indicators of water quality and the sources of water contamination together with the local stakeholders. By understanding the source–impact relation, an IWRM policy framework can be better directed to the most polluted areas in the most critical periods. As these practices are developed together with the local stakeholders, they are aligned with the local and national context. The articulation of this commitment stimulates important stakeholders (such as the International Development Bank, the Ministry of Water and Environment and other governmental departments, local NGOs, and others) to become actively involved in the project with the UCB.

What can be done to overcome the barriers for the improvement of IWRM policies to respond to the needs of the different stakeholders, and especially of the most vulnerable social groups in the Katari River Basin? Certainly, more "technical" information is still needed about water quantity and quality. The UCB water specialists can help identify the main sources and pollutants causing the deterioration of water quality in the main rivers of the Katari sub-basin. They can also help clarify the relationship between these sources and their impact on water quality, and establish the required frequency of monitoring for the proper assessment of water quality.

However, this information will not suffice to deal with the intricate problems in the Katari River Basin. That implies a better understanding of the different water management policies that were developed by the national, regional, and local authorities to face the multiple problems of the Katari sub-basin: how were they developed and implemented, and how were different actors involved or excluded from this process? Currently, a thorough review and analysis of the existing documents related to the IWRM policies of the KRB (river basin plans, government reports, environmental publications, formal regulations, and scientific publications) is complemented by field work featuring in-depth interviews with all stakeholders related to the Katari basin, including representatives of national, regional, and local authorities, managers of industrial companies, local water utilities and water cooperatives, farmer's associations, community leaders, and local water activists. This information helps outline the diverse activities that take place along the river basin and understand how they influence or are impacted by the water quality deterioration of the river.

Talking with all these stakeholders not only enhances the understanding of how the integrated water resource policy was developed and implemented, it directly generates new ideas about the role of the local actors in this process. Interviewees are stimulated to reflect on their interests, roles, and responsibilities. By feeding their ideas back to the other stakeholders, the research is stimulating a joint reflection and co-creation process for the IMKRB, in which the concerns of a vulnerable environment and of the most vulnerable social groups are included.

## 6. Conclusions

Integrated Water Resources Management was once conceived as a panacea to address water problems by coordination between different governmental, agencies, policy domains, and administrative levels. Later, it became clear that the wicked nature of water-related problems requires the active incorporation of a variety of stakeholders from the private and civil sector in collaborative water governance. Vulnerable social groups in local communities deserve special attention in this governance process. They often have important contextual information that should be taken into account and they are confronted most directly with the consequences of inadequate river basin management, as the KRB clearly illustrated.

In this publication, we draw attention to the implications of the shift toward collaborative governance as a way of doing research. All of the relevant stakeholders should be involved throughout the whole research process, rather than only in the last phase, when conclusions are translated into new policies. TLCs have to come up with research questions that include and reflect the worries of the most vulnerable social groups. TLCs are also mechanisms for joint information gathering and analysis. They offer spaces in which formerly excluded social groups can express how they make sense of the situations and problems with which they are confronted. Together with them, we have to reflect on how can they be empowered to be actively involved in the IWRM and in this way play a proactive role in the improvement of their own reality.

In our research practice, we are confronted with the challenge of connecting scientific and local ways of knowing, because science aims at producing generalizable knowledge with objectifying, distant methods, whereas local community knowledge aims at practical and context-specific solutions that are rooted in the subjective and rich experience of intimate and engaged contact with the environment. Overcoming these epistemological differences and incongruences is complicated because of the unequal prestige and power between academic and local knowledge holders. The TLC approach of the UCB and VLIR-UOS cooperation program offers an opportunity for academic researchers and external stakeholders, including the most vulnerable groups, to learn by doing and experimenting with joint practices that contribute to the co-creation of scientifically sound, locally adequate, and socially relevant water policies.

In this way, our research contributes to the development of a transdisciplinary approach for the design and implementation of IWRM policies, taking into account the particularities of the local geographical context, the governance structure, the institutional framework, and the complex nature of the water sector. Follow-up research will determine the long-term impact of this innovative approach on water quality and quantity, from the perspective of the different stakeholders.

**Author Contributions:** Conceptualization, M.C. and A.A.; Methodology, M.C.; Formal Analysis, M.C. and A.A.; Investigation, M.H., A.A., M.C. and M.B.; Resources, M.H., A.A. and M.C.; Data Curation, M.C. and A.A.; Writing-Original Draft Preparation, M.C. and A.A.; Writing-Review & Editing, M.H., A.A. and M.C.; Visualization, M.C. and A.A.; Supervision, M.C. and A.A.; Project Administration, M.C. and A.A.

**Funding:** This research is funded by the VLIR-UOS, IUC 2017 Phase 1 UCB-B https://www.vliruos.be/en/projects/project/22?pid=3607.

**Acknowledgments:** The authors acknowledge the support of the local communities of the Municipality of Pucarani and the Unidad Gestora de la Cuenca Katari for their support providing valuable information.

**Conflicts of Interest:** The authors declare no conflict of interest.

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
