# Peer review of "Transdisciplinary Learning Communities to Involve Vulnerable Social Groups in Solving Complex Water-Related Problems in Bolivia"

_water, doi:10.3390/w11020385_

Round 1
Reviewer 1 Report
The article is interesting and provides input to scientific research on Bolivia water aspects which are complex and small peer-review literature is available. However, the article needs strong English editing. In particular, with the use of adjectives that do not apply to the type of language expected in a scientific journal, such as the word "fuzzy". Also, there is an overuse of "connecting words" that are misplaced, in particular when starting a new paragraph.
The article needs to improve on referencing the statements made. For example, the first paragraph of the article contains a lot of statements with no reference offered.
Methodology section: needs to be improved. It needs to clearly explain and describe the steps taken so other researchers could potentially replicate the research.
The case study description could be part of the introduction or the results.
I'm having a hard time understanding the discussion session. It seems that it should be the results session instead of the discussion session since the authors are offering the answer to their research questions. I would suggest making the discussion session, the results session.
The authors need to summarize the answers to the questions and summarize the key insights of those results. This summary is your discussion and conclusions session.
Author Response
Point 1.- The article is interesting and provides input to scientific research on Bolivia water aspects which are complex and small peer-review literature is available. However, the article needs strong English editing. In particular, with the use of adjectives that do not apply to the type of language expected in a scientific journal, such as the word "fuzzy". Also, there is an overuse of "connecting words" that are misplaced, in particular when starting a new paragraph.
Response 1: An extensive English review of the document was performed.
Point 2: The article needs to improve on referencing the statements made. For example, the first paragraph of the article contains a lot of statements with no reference offered.
Response 2: Additional references were incorporated to the manuscript paying special attention to the first section.
Point 3: Methodology section: needs to be improved. It needs to clearly explain and describe the steps taken so other researchers could potentially replicate the research.
Response 3: As we are reporting about an ongoing research, it is not possible yet to describe all steps that were taken, and we explain the principles and criteria that justify the choices that we are currently making. The readers are warned that action research – unlike experimental research – cannot just be replicated, as the course and outcomes depend to a large degree on the specific context, opportunities and involved actors. A “thick description” of this context and actors must allow to adapt the learning conclusions of this case to other (similar) cases.
Point 4: The case study description could be part of the introduction or the results.
Response 4: The case study description is now presented in the results section.
Point 5: I'm having a hard time understanding the discussion session. It seems that it should be the results session instead of the discussion session since the authors are offering the answer to their research questions. I would suggest making the discussion session, the results session.
Response 5: We have restructured the manuscript, splitting between a (short) discussion section and Conclusion section. In the discussion section we reflect briefly on the challenges with which we are confronted applying the TLC approach in the Katari river basin. In the last conclusion section we synthesize and clarify the intention and limitations of this article: a description of the opportunities offered by an alternative transdisciplinary approach, currently put in practice in the interesting Bolivian context.
Point 6: The authors need to summarize the answers to the questions and summarize the key insights of those results. This summary is your discussion and conclusions session.
Response 6: cfr. Reaction to 5
Reviewer 2 Report
Journal Water
Manuscript ID : water-407880
Title: Transdisciplinary Learning Communities to involve vulnerable social groups in solving complex water-related problems in Bolivia.
Dear Authors,
I would like to congratulate you all for such an interesting paper. I think it addresses an issue worldwide important and regional critical.
My comments are related to the scientifically soundness of the paper.
· The whole section 1, lacks of References (eg. line 30-38, 39-40; 44-46).
· Besides mentioning the MDG´s, please consider a discussion on the Sustainable Development Goals (SDG´s).
· Use acronyms (eg WHO; line 33)
· Edit: “become a crucial element” (line 42)
· Section 2: In this section, different concepts are mentioned, but have not being sufficiently theorized. Eg. IWRM (52); Sustainable Development (65), Water Governance (69-70); Equity (59)
· Besides, the authors forget that the unit for the IWRM is the basin which integrates surface water resources and groundwater resources; not only River as the authors think (line 66).
· Something that strikes me as particularly important is the lack of theoretical discussion in order to conclude that IWRM is the best water management approach. In other words, the authors do not take the time to confront authors, points of view, related to this approach and others. If this is the “best” water management approach it is because the authors believe so, of because they have concluded it from a study? In this case, I would be very interested in knowing more about how, you the authors, have come to this conclusion. Please provide a theoretical thinking to support your affirmative, and definitive, statements.
· One more thing on this section: Do you think the relation between IWRM and Water Governance is linear, the first one leads to the second one? Please explore further this question, and provide a theoretical reference.
· Section 3, pretty much the same!
· Concepts lacking scientific support: Sustainable Natural Resources Management (95-96); proper water governance (and improper?); Sustainability of Water Resources (103); Sustainable Water Resources Management (119).
· Line 110: Non, potential water conflicts are not evident!
· Section 4: Collaborative modes of governance (161); Multi-actor governance (163); Collaborative Governance (188).
I have focused my comments to this paper on the ensemble of terms and-or concepts because the authors use them indistinctly and without providing a sufficient scientific and theoretical support. I invite them to think about these concepts and to find their own theoretical thinking taking as a reference what has been already said about each of these concepts. Discuss your references, and make your own conclusions! I think this exercise will help you to further, and better, sustain the important approach adopted by the UCB.
Author Response
Point 1
· The whole section 1, lacks of References (eg. line 30-38, 39-40; 44-46).
· Besides mentioning the MDG´s, please consider a discussion on the Sustainable Development Goals (SDG´s).
· Use acronyms (eg WHO; line 33)
· Edit: “become a crucial element” (line 42)
Response 1:
Additional references were incorporated paying special attention to the first section
The acronym WHO is inserted
“Become a crucial element” was incorporated.
Point 2:
· Section 2: In this section, different concepts are mentioned, but have not being sufficiently theorized. Eg. IWRM (52); Sustainable Development (65), Water Governance (69-70); Equity (59)
· Besides, the authors forget that the unit for the IWRM is the basin which integrates surface water resources and groundwater resources; not only River as the authors think (line 66).
· Something that strikes me as particularly important is the lack of theoretical discussion in order to conclude that IWRM is the best water management approach. In other words, the authors do not take the time to confront authors, points of view, related to this approach and others. If this is the “best” water management approach it is because the authors believe so, of because they have concluded it from a study? In this case, I would be very interested in knowing more about how, you the authors, have come to this conclusion. Please provide a theoretical thinking to support your affirmative, and definitive, statements.
· One more thing on this section: Do you think the relation between IWRM and Water Governance is linear, the first one leads to the second one? Please explore further this question, and provide a theoretical reference.
Response 2:
The concepts employed in the manuscript were better theorized and theoretical references where incorporated to the document.
A paragraph was incorporated to discuss the river basin as the management unit recalled at the IWRM approach.
A theoretical discussion was incorporated to support the recognition of the IWRM approach as the best water resource management approach
Point 3.-
· Section 3, pretty much the same!
· Concepts lacking scientific support: Sustainable Natural Resources Management (95-96); proper water governance (and improper?); Sustainability of Water Resources (103); Sustainable Water Resources Management (119).
· Line 110: Non, potential water conflicts are not evident!
Response 3:
Concepts were developed in section 3 and we also incorporated the references to support them.
The water conflict sentence was edited.
· Section 4: Collaborative modes of governance (161); Multi-actor governance (163); Collaborative Governance (188).
Concepts were developed in section 4 and we also incorporated the references to support them.
I have focused my comments to this paper on the ensemble of terms and-or concepts because the authors use them indistinctly and without providing a sufficient scientific and theoretical support. I invite them to think about these concepts and to find their own theoretical thinking taking as a reference what has been already said about each of these concepts. Discuss your references, and make your own conclusions! I think this exercise will help you to further, and better, sustain the important approach adopted by the UCB.
We thank the reviewer for the insightful comments that helped us making adaptations to improve the manuscript.
Round 2
Reviewer 1 Report
Some minor English editing is still needed. The paper looks much better after the changes made.
Author Response
Dear Reviewer,
The authors appreciate your feedback which allowed us to improve the manuscript. We edited the document to improve the quality and its scientific soundness. Following the main edits related to your last review:
Point 1.-
Open Review
(x) I would not like to sign my review report
( ) I would like to sign my review report
English language and style
( ) Extensive editing of English language and style required
( ) Moderate English changes required
(x) English language and style are fine/minor spell check required
( ) I don't feel qualified to judge about the English language and style
Response 1.-
Some minor language and spelling errors have been corrected
Point 2.-
Is the research design appropriate?
( )
(x)
( )
( )
Are the methods adequately described?
( )
(x)
( )
( )
Response 2.-
The methods section has been re-structured and partly re-written
Reviewer 2 Report
Journal Water
Manuscript ID water-407880
Title : Transdisciplinary Learning Communities to involve vulnerable social groups in solving complex water-related problems in Bolivia.
Despite the fact that the authors have greatly improved the quality of the manuscript, still, some methodological and content issues need to be solved.
Major revision
1. One of the biggest challenges of the manuscript consist on the fact that the authors don’t make any difference between the Methods of the paper and the Transdisciplinary Learning Community (TLC) approach. The section results, discussion and conclusion lack substance.
2. The TLC, and its application on the Katari river basin, represent the case study of the manuscript.
3. In fact, the Introduction of the manuscript should clearly state the following:
a. A robust literature review on the concepts the authors will address further (sections 2 and 3 should be integrated).
b. The introduction section should also mention the research questions, the organization of the paper, and the purposes the authors are looking for through the TLC and its application on the Katari river basin.
c. Some theoretical notions that should be mentioned, and justified, in the introduction section: IWRM, Water governance, sustainable natural resources management, water wars, sustainable water resources management, collaborative governance, and wicked problems.
d. Again, the authors use indistinctly these notions and do not provide a strong theoretical framework. Eg, could you explain the differences between the sustainable natural resources management and sustainable water resources management?
4. The manuscript, in its present form, lacks of a Materials and Methods section.
a. This issue could be solved once the authors are able to reformulate its purposes and revise the introduction section.
b. Note that the TLC approach does not represent the Methodology of the paper!!! This approach, and its application on the Katari river basin, is the case study of the paper. It is not the Method of the manuscript.
5. A section on the Case study should address the TLC and the Katari river basin.
a. Eg. Line 245-272 is about the case study!!
b. Section 5 is all about the case study.
6. The section 6 Results does not represent the results of the manuscript!!!
a. This section is all about the case study.
b. Eg. Line 379 – 386 : Do you really consider the description of the Katari river basin is a result from your research??? The information provided here describes the case study.
c. Idem 389-402. This information does not result from your research work!
7. Section 5 discussion: Why the discussion section include research questions? Do not you think these questions should be mentioned before!?!
8. Section 6 conclusion: I am now confused. How many questions are addressed in the whole paper?
Specific comments:
Line 64 mention the date of the event
Line 94 “IWRM must find a balance” ….are you sure? Is this what you want to say? I suggest to reformulate your idea.
Line 106 United States
Line 106-109 please provide river basin examples! And be sure about the “implementation in the US”
Line 120-121 please provide references
Line 145 “production in various sectors” provide examples and explain further.
Line 144-146 please provide references
Line 318-320 this is important information for the conclusions sections. As you can notice, this argument is related to the relation between the TLC and the IWRM approaches.
Line 327-328 please provide references.
Line 340-344 please provide references
Line 371-377 this is a mix of results and conclusions
Author Response
Dear reviewer,
Thanks for your critical remarks and constructive suggestions that have helped us to continue improving our manuscript. We have added, adapted and moved certain parts throughout the whole manuscript, to address your comments. We hope you can agree with these changes.
Sincerely
The authors
Major revision
1. One of the biggest challenges of the manuscript consist on the fact that the authors don’t make any difference between the Methods of the paper and the Transdisciplinary Learning Community (TLC) approach. The section results, discussion and conclusion lack substance.
Reaction: Action research is the methodology used for the research underlying this manuscript. In action research data gathering and analysis on the one hand and contributing to improvements in the study case (in this case with TLC) are done simultaneously and need each other. That is the reason why we included the TLC approach in the Katari basin in the methodology section (3), explaining and justifying this inclusion. The more conceptual grounding of transdisciplinarity is now included in the conceptual section 2 on IWRM and collaborative governance.
2. The TLC, and its application on the Katari river basin, represent the case study of the manuscript.
Reaction: Indeed, we consider what we learned with the TLC approach about the water governance in Bolivia and how this affects the water governance challenges in the Katari river basin, as the result of the case study. We state this more clearly in this version of the manuscript
3. In fact, the Introduction of the manuscript should clearly state the following:
a. A robust literature review on the concepts the authors will address further (sections 2 and 3 should be integrated).
Reaction: as explained above, the more theoretical explanations of section 3 are now integrated in section 2, and the literature review on the main concepts has been extended.
b. The introduction section should also mention the research questions, the organization of the paper, and the purposes the authors are looking for through the TLC and its application on the Katari river basin.
Reaction: we recognize that this was still lacking and we have added these parts to the introduction section 1.
c. Some theoretical notions that should be mentioned, and justified, in the introduction section: IWRM, Water governance, sustainable natural resources management, water wars, sustainable water resources management, collaborative governance, and wicked problems.
Reaction: we have now introduced and explained the main theoretical notions in the introduction section 1
d. Again, the authors use indistinctly these notions and do not provide a strong theoretical framework. Eg, could you explain the differences between the sustainable natural resources management and sustainable water resources management?
Reaction: we consider water resources management as a specific but typical example of natural resources management (that includes as well other natural resources, like soils, fauna and flora, etc.)
3. The manuscript, in its present form, lacks of a Materials and Methods section.
Reaction: we have restructured and re-written the methods section (3). We are also clear and transparent that this manuscript is about an on-going action research. That is the reason that we can report about the challenges it raises currently for collaborative governance, but not (yet) about the impact of this collaborative governance on water quantity and quality and/or on the living conditions of the vulnerable communities in the area.
a. This issue could be solved once the authors are able to reformulate its purposes and revise the introduction section.
Reaction: We hope the reformulation satisfies the reviewer.
b. Note that the TLC approach does not represent the Methodology of the paper!!! This approach, and its application on the Katari river basin, is the case study of the paper. It is not the Method of the manuscript.
Reaction: The TLC approach is part of the action research methodology (explained above). This is now explained more clearly in the manuscript.
4. A section on the Case study should address the TLC and the Katari river basin.
Reaction: The factual (geographical) information about the Katari river basin is included in the methodological section, to ground the selection of this river basin as a convenient case for a study about the challenges of collaborative water governance in the Andes.
a. Eg. Line 245-272 is about the case study!!
Reaction: This part is now moved to the new section 3 (TLC as action research methodology) in which we explain the application of the TLC approach at the UCB in Bolivia.
b. Section 5 is all about the case study.
Reaction: section 5 has been completely re-structured. Some parts have been incorporated in section 2 (theoretical notions, other parts in section 3 about TLC as action research, and still other parts are incorporated in section 4 (results) and 5 (conclusions) as suggested by the reviewer.
5. The section 6 Results does not represent the results of the manuscript!!!
Reaction: we are now clearer about what we consider the results of our study, in an introductory paragraph, as follows: “We start this section with an analysis of how water policies have evolved in Bolivia and how this has impacted socio-ecological conditions in the river basins in recent history, and then we analyze more in-depth the challenges for collaborative water governance in the Katari river basin.”
a. This section is all about the case study.
b. Eg. Line 379 – 386 : Do you really consider the description of the Katari river basin is a result from your research??? The information provided here describes the case study.
Reaction: that is the reason that we have moved this information to section 3, which describes how (and where) the case study is put in practice – see above
c. Idem 389-402. This information does not result from your research work!
Reaction: indeed, we changed this information to section 3
6. Section 5 discussion: Why the discussion section include research questions? Do not you think these questions should be mentioned before!?!
Reaction: we removed these questions from the conclusions section and we included some of them in the introduction section.
7. Section 6 conclusion: I am now confused. How many questions are addressed in the whole paper?
Reaction: there is one central question, presented as such in the introduction
Specific comments:
Line 64 mention the date of the event
(done)
Line 94 “IWRM must find a balance” ….are you sure? Is this what you want to say? I suggest to reformulate your idea.
We understand the possible misunderstanding and we reformulated this part as follows: “IWRM must take into account economic feasability, social equity and environmental issues.. Moreover, since the time of origin of that definition, more emphasis has been put on ecological boundaries and social inclusiveness [15]…”
Line 106 United States
(done)
Line 106-109 please provide river basin examples! And be sure about the “implementation in the US”
(done)
Line 120-121 please provide references
(done)
Line 145 “production in various sectors” provide examples and explain further.
(done)
Line 144-146 please provide references
(done)
Line 318-320 this is important information for the conclusions sections. As you can notice, this argument is related to the relation between the TLC and the IWRM approaches.
The message in now part of the conclusions section
Line 327-328 please provide references.
(done)
Line 340-344 please provide references
(done)
Line 371-377 this is a mix of results and conclusions
part of it is moved to the conclusions section
Round 3
Reviewer 2 Report
Journal Water (ISSN 2073-4441)
Manuscript ID: water-407880
Title: Transdisciplinary Learning Communities to involve vulnerable social groups in solving complex water-related problems in Bolivia.
Dear Authors,
I would like to congratulate you all for the very interesting, precise and valuable research efforts you have done in order to improve the scientific quality of the manuscript.
Still, some very small typos to edit, please:
Line 52: “Scientists and social scientists”
Suggestions: a transdisciplinary group of scientists, or as you mention on line 66 “between scientists of different disciplines”, or on line 579 “researchers from different disciplines”
line 70: delete huge
line 75 : I suggest an acronym that you can use afterwards : Katari River Basin (KRB).
Line 80: Be careful, you have already used the acronym for of Integrated Water Resource Management
Line 694-699: Try, please to integrate all these ideas. I guess you can write a more concise and integrated sentence-s.
Line 1049: Could you please mention his name?
Line 1182: I suggest you use an acronym for “ integrated management of the Katari river” (….IMKRB).that you can use afterwards.
Line 1232: use the acronym
Line 1261-1262: use the acronym: if you agree.
Once more, congratulations for your research efforts.
Author Response
Dear reviewer,
The authors appreciate your feedback to improve the manuscript. Following we answer to your last review:
Line 52: “Scientists and social scientists”
Suggestions: a transdisciplinary group of scientists, or as you mention on line 66 “between scientists of different disciplines”, or on line 579 “researchers from different disciplines”
Response. - We replaced “scientist and social scientist” for “scientist of different disciplines”
line 70: delete huge
Response. - Done
line 75 : I suggest an acronym that you can use afterwards : Katari River Basin (KRB).
Response. - The acronym is now incorporated in the document
Line 80: Be careful, you have already used the acronym for of Integrated Water Resource Management
Response. - Done
Line 694-699: Try, please to integrate all these ideas. I guess you can write a more concise and integrated sentence-s.
Response. - The sentence has been edited to integrate the ideas
Line 1049: Could you please mention his name?
Response. - The name is now mentioned
Line 1182: I suggest you use an acronym for “ integrated management of the Katari river” (….IMKRB).that you can use afterwards.
Response. - The acronym is now incorporated in the document
Line 1232: use the acronym
Response. - Done
Line 1261-1262: use the acronym: if you agree
Response. - Done